# A Comprehensive Review on Dietary Polysaccharides as Prebiotics, Synbiotics, and Postbiotics in Infant Formula and Their Influences on Gut Microbiota

**DOI:** 10.3390/nu16234122

**Published:** 2024-11-28

**Authors:** Wenyuan Zhang, Yanli Zhang, Yaqi Zhao, Liang Li, Zhanquan Zhang, Kasper Hettinga, Haixia Yang, Jianjun Deng

**Affiliations:** 1State Key Laboratory of Vegetable Biobreeding, Institute of Vegetables and Flowers, Chinese Academy of Agricultural Sciences, Beijing 100081, China; zhangwenyuan@caas.cn (W.Z.); 13934604283@163.com (Y.Z.); zhaoyaqii@163.com (Y.Z.); scy_ll@163.com (L.L.); zhangzhanquan@caas.cn (Z.Z.); 2Dairy Science and Technology, Food Quality and Design Group, Wageningen University & Research, 6708 WG Wageningen, The Netherlands; kasper.hettinga@wur.nl; 3College of Food Science and Nutritional Engineering, China Agricultural University, Beijing 100083, China; hyang@cau.edu.cn

**Keywords:** dietary polysaccharides, prebiotics, postbiotics, gut microbiota, infant formula

## Abstract

Human milk contains an abundance of nutrients which benefit the development and growth of infants. However, infant formula has to be used when breastfeeding is not possible. The large differences between human milk and infant formula in prebiotics lead to the suboptimal intestinal health of infant formula-fed infants. This functional deficit of infant formula may be overcome through other dietary polysaccharides that have been characterized. The aim of this review was to summarize the potential applications of dietary polysaccharides as prebiotics, synbiotics, and postbiotics in infant formula to better mimic the functionality of human milk prebiotics for infant gut health. Previous studies have demonstrated the influences of dietary polysaccharides on gut microbiota, SCFA production, and immune system development. Compared to prebiotics, synbiotics and postbiotics showed better application potential in shaping the gut microbiota, the prevention of pathogen infections, and the development of the immune system. Moreover, the safety issues for biotics still require more clinical trials with a large-scale population and long time duration, and the generally accepted regulations are important to regulate related products. Pectin polysaccharides has similar impacts to human milk oligosaccharides on gut microbiota and the repairing of a damaged gut barrier, with similar functions also being observed for inulin and β-glucan. Prebiotics as an encapsulation material combined with probiotics and postbiotics showed better potential applications compared to traditional material in infant formula.

## 1. Introduction

Human milk is considered as the most natural food source for infants. It contains a variety of components such as proteins, lipids, minerals, and carbohydrates, which can provide both short- and long-term benefits for infants. Carbohydrates are the major component in human milk, contributing a total of 59% dry matter to human milk. Carbohydrates in human milk mainly consist of lactose and other sugar-conjugated compounds, such as free oligosaccharides. The concentrations of human milk oligosaccharides (HMOs) are up to 25 g/L in colostrum and from 10 to 15 g/L in mature milk. The beneficial effects of HMOs have been reported to involve gut microbiota, antiadhesive antimicrobials, intestinal epithelial cell modulators, immune modulators, protection against some diseases, and brain development [1]. This is due to the indigestible properties of HMOs by gastric acidity, host enzymes, and gastrointestinal absorption (~1%). For example, genome results showed that *Bifidobacterium longum* subsp. *infantis* can encode specific glycosidases, sugar transporters, and glycan-binding proteins [2], and thus utilize HMOs to produce mono- and disaccharides for their growth.

Over the past several decades, the production of infant formulas has continuously been optimized to make it more closely resemble human milk. Although there are numerous beneficial effects for infants of human milk, infant formula has to be used when breastfeeding is not possible. Infant formula should be designed to meet the nutritional requirements and mimic the biological functions for non-breastfed infants. Commercial infant formula is normally animal milk-based, such as bovine or goat milk. However, the concentration of oligosaccharides in bovine milk and in goat milk is much lower than that in human milk. Moreover, the oligosaccharide profiles in animal milks have large differences to human milk. Human milk contains higher amounts of neutral and fucosylated oligosaccharides, while it is lower in sialylated oligosaccharides. In contrast, the sialylated oligosaccharides are the dominant ones in bovine milk, with relatively lower contents of neutral and fucosylated oligosaccharides [3]. This reveals that there are large differences between human milk and infant formula, which may result in different biological functions for infants. To better mimic the prebiotic functions of human milk, several polysaccharides are added into infant formula as prebiotics such as galacto-oligosaccharides (GOSs) and fructo-oligosaccharides (FOSs). However, these prebiotics in infant formula present distinct activity to HMOs, for example, regarding the infant gut microbiota development during their early life. This is mainly due to the different effects of HMOs and GOSs/FOSs on the growth of infant gut microorganisms.

Dietary polysaccharides refer to a type of macromolecule consisting of more than ten monosaccharide units and normally are the major components of dietary fiber from many food products [4]. Many studies provide more in-depth information about dietary polysaccharides and their biological functions, especially as prebiotics and their benefits on gut microbiota. However, their applications in infant formula production compared to normal formula combined with probiotics and/or postbiotics still remain unknown, especially regarding short- and long-term effects on infants. Therefore, this review focuses on the different effects of biotics in infant formula and human milk on gut microbiota for infants and discusses studies about the biological functions of dietary polysaccharides to explore the possibility to add them to infant formula. Moreover, future perspectives on polysaccharides as synbiotics/postbiotics used for infant formula development will be discussed.

## 2. Challenges for Prebiotics in Infant Formula

An overview of gut microbiota development between infants fed human milk versus infant formula is shown in Figure 1. This figure clearly shows that the gut microbiota development between these infants is significantly different. However, the main reason for the differences in infant gut microbiota developed is the difference in biological functions between HMOs in human milk and prebiotics in infant formula. The gut microbiota of healthy breastfed infants is dominated by a number of different *Bifidobacterium* species such as *B. breve*, *B. bifidum*, and *B. infantis.* Compared to breastfed infants, infants fed with infant formula harbor less diverse *Bifidobacterium* microbiota, being more often colonized by other microorganisms (Figure 1). For example, some potential opportunistic pathogens such as *Clostridium difficile*, *Clostridium perfringens*, *enteropathogenic Escherichia coli*, and *enteroaggregative Escherichia coli* are often found in infant formula-fed infants’ guts [5]. The gut microbiota in the formula-fed infants have higher diversity, which is more likely in older children. The different compositions of prebiotics and their effects on gut microbiota lead to several potential risks for infant formula-fed infants. For instance, a study reported that compared to infant formula feeding (30%), the prevalence of *C. difficile* was significantly lower in breastfed infants (14%) [6]. Therefore, preventing this infection, for example, through breastfeeding, may increase the infant’s health. Moreover, the use of antibiotics in infants has been reported to cause a decrease in beneficial bacteria, increase in harmful bacteria, and a loss of diversity in the gut flora [7]. For example, a randomized trial showed that antibiotic-treated infants had less abundance of *Bifidobacterium* spp., but an increasing abundance of *Klebsiella* and *Enterococcus* spp. [8]. Breastfeeding may reduce the disruptions to the infant gut microbiome after antibiotic treatment [9].

Only a well-developed gut microbiota such as in breastfed infants can provide health benefits to the host, and the dominant benefits, as shown in Figure 1, include (1) more resistance to pathogen infections, (2) proper immune system development, (3) supporting gut barrier development, and (4) brain and neurocognitive development. The diversity of gut microbiota is associated with immune system development in breastfed infants, and formula-fed infants may lead to variations in their immune system and related diseases. These health benefits from gut microbiota are mainly contributed by the fermentation of HMOs or prebiotics to produce a variety of metabolites, such as SCFAs. SCFAs stimulate the G protein-coupled receptors (GPRs) on intestinal epithelial cells (IECs) and activate downstream cell signal transduction to regulate the immune response of T cells, B cells and dendritic cells [10]. The SCFAs impact the differentiation of T cells into various types of Treg cells and Th cells, and thereby induce the development of immune tolerance via anti-inflammatory cytokines such as *Foxp3* and IL-10 [11]. For example, *Foxp3* is involved in the establishment and maintenance of the Treg cell phenotype and inhibits the activation of inflammatory cytokines such as NF-κB [12]. Furthermore, *Bifidobacterium* species can convert aromatic amino acids (tryptophan, phenylalanine, and tyrosine) and produce their respective aromatic lactic acids via aromatic lactate dehydrogenase [13]. These aromatic lactic acids as aryl hydrocarbon receptor (AhR) ligands activate AhR signaling to control gut homoeostasis, gut barrier integrity, and immune responses via the regulation of many immune cell types such as intraepithelial lymphocytes, CD4 + T cells, monocytes, and Th17 cells [13,14]. These metabolites from gut microbiota stimulate the immunity cells, which in turn affect the gut barrier-mediated intestinal homeostasis. The gut barrier is composed of epithelial cells, goblet cells, Paneth cells, stem cells, tuft cells, and enteroendocrine cells connected by an apical junctional complex such as claudins, occludin, adherence junction proteins, and desmosomes [15]. The newborn at birth has an underdeveloped intestinal immune system and a highly permeable gut barrier for the better absorption of nutrients, whereas the gut barrier decreases its permeability during their first two years of life. This leads to increased risks of microbial translocation and inflammatory gastrointestinal diseases, such as necrotizing enterocolitis (NEC), in early life, especially in preterm-born infants. In early life, the gut microbiota plays critical roles in the development of the gut barrier, including the development of the mucus layer and intestinal epithelial barrier [16]. Moreover, the gut microbiota also impacts the endocrine pathway, the brain, and the neurocognitive development of infants [17,18].

Many studies reported that the colonization of gut microbiota not only impacted the host’s immunity, but was also associated with the onset of autoimmune or allergic diseases and prevented disease later in life. For example, the higher abundance of several probiotics such as *Lactobacillus*, *Bifidobacterium*, and *Akkermansia* have been reported to reduce the risk of allergy. In contrast, the colonization of some harmful pathogens including *Staphyloccocus aureus*, *C. difficile*, and *Rhodotorula* in the gut was associated with a higher risk of developing an allergy later in life [19]. This is mainly caused by the changes in SCFA production, especially of butyrate and acetate, which have been reported to be associated with immune system development [20]. SCFAs such as acetate, butyrate, and propionate, have been reported to stimulate T cell proliferation and reduce the immune responses by allergy-associated Th2 cells [21]. Overall, the infant gut microbiota constitutions show large differences among different feeding practices due to the oligosaccharides and other bioactive compounds in human milk and infant formula, which impact the health benefits for infants for the short and long term.

## 3. Prebiotics, Synbiotics, and Postbiotics

HMOs are a unique component in human milk with more than 200 structures of HMOs, while the function and structure of all HMOs are still hardly studied and limit the development of prebiotics supplements in infant formula. Although infant formula ingredients have been developed for about 200 years, its composition is still different from human milk. One specific difference of interest is regarding the oligosaccharides. In this section, different ingredients (prebiotics, synbiotics, postbiotics) will be discussed regarding their functional requirements and their potential application in infant formula.

### 3.1. Prebiotics

#### 3.1.1. Specific HMOs in Infant Formula

For now, only 2-fucosyllactose (2′-FL) and lacto-N-neotetraose (LNnT) are commercially allowed to be added into infant formula as prebiotics. LNnT and 2′-FL have been shown to benefit the growth of *Bifidobacterium*, inhibit the growth of pathogens, stimulate the gut intestinal barrier, and support the central nervous system [22]. The production of HMOs by cell factories as well as their safe use in infant formula still needs more in vivo and clinical experiments, to study their short- and long-term health effects.

#### 3.1.2. Oligosaccharides in Animal Milks

The basic reason for the prebiotics supplement in infant formula is due to structural differences between the oligosaccharide constitution of bovine milk and HMOs. Apart from bovine milk, there are a few studies on other animal milk sources, such as goat milk and camel milk [23]. An *in vivo* study demonstrated that, compared to bovine milk infant formula, mice fed with goat milk infant formula had a more similar gut microbiota diversity to breastfed infants [24]. In summary, other animal milks, especially goat milk infant formula, may have better health influences on infants compared to normal formula, while the related consequences on the health benefits for infants still require more clinical experiments in the future.

#### 3.1.3. Galacto-Oligosaccharides (GOSs) and Fructo-Oligosaccharides (FOSs)

GOSs are the intermediate products from enzymatic lactose hydrolysis and are normally composed by glucose and multiple galactose units with diverse glyosidic bonds such as β-(1-3), β-(1-4), and β-(1-6) [25]. The degree of polymerization (DP) of GOSs ranges from 2 to 8. Various studies have reported that GOSs in infant formula can benefit the growth of beneficial bacteria such as special *Bifidobacterium* species, which reduce the colonization of pathogens, resulting in a more similar intestinal microbiota composition to that in breastfed infants [26]. The fermentation of GOSs by gut microbiota results in the production of SCFAs and other metabolites, which can influence the immune system in the gastrointestinal tract. Furthermore, there are a few studies that have demonstrated other biological functions of GOSs such as the prevention of pathogen adhesion and effects on gut barrier function [27]. An FOS is mainly composed of a small number of fructose units connected with (2-1)-β-glycosidic bonds and a single D-glucosyl unit at the non-reducing end. Normally, the average DP used in infant formula is more than 23, and this is produced by the hydrolysis of inulin [28]. The biological function of FOSs is due to their non-digestible property and their ability to reach the intestine intact, acting as prebiotics to benefit the growth of gut microbiota, especially *Bifidobacterium* and *lactobacillus* species [29].

In the last few decades, different mixtures of GOSs/FOSs have been investigated. Recently, the most studied prebiotics mixture in infant formula combines short-chain (sc) GOSs and long-chain (lc) FOSs in a ratio of 9:1, as this was shown to better mimic the molecular size distribution of HMOs [27]. The concentration of the scGOS/lcFOS mixture is usually around 8 g/L, with more than 100 different structures, being in the range of the HMO concentration and diversity in human milk [22]. Various clinical studies on scGOSs/lcFOSs (9:1) have demonstrated benefits on the growth of gut microbiota, modulation of the immune system, reduction in infections by pathogens, and stool softening, targeting especially high-risk infants [30]. Several randomized controlled double-blind clinical studies showed that this formula prevented atopic disease in at-risk infants. Besides that, the long-term influences of this formula have been widely reported for infants of 6 months to 5 years [31].

### 3.2. Synbiotics

Synbiotics refer to a combination of probiotics and prebiotics to selectively increase the abundance of beneficial microbes in the infant gut. Human milk also contains 10^3^ to 10^5^ cfu/mL beneficial microbiota. Several studies have reported numerous health benefits of synbiotics such as the regulation of gut microbiota in infants [32], sleep and settling behaviors [33], and reduced systemic inflammation [34]. For example, a randomized and double-blind study studied the gut microbiota in 290 infants (6–19 weeks) who received infant formula with synbiotics containing infant-type *Bifidobacterium* strain *B. breve* (*M-16V)* (1 × 10^4^ cfu/mL or 1 × 10^6^ cfu/mL) with 0.8 g/100 mL scGOSs/lcFOSs (9:1) compared with breastfed infants as a reference [5]. Their results showed that after 6 weeks of intervention, supplementation with synbiotics at two different doses both increased the *Bifidobacterium* proportions in healthy infants and decreased the abundance of potential pathogens such as *C. difficile*. This infant formula supplemented with synbiotics resulted in a lower fecal pH and significant changes in short-chain fatty acid composition, with a higher proportion of acetate and less butyric, isobutyric, and isovaleric acids [35]. Overall, this indicates the possibility of using synbiotics (including FOSs/GOSs/*B. infantis*) for infant formula specifically targeting specific groups, such as infants delivered by cesarean section and infants with an increased risk of allergy. However, the long-term effects of infant formula supplemented with synbiotics still need to be investigated and compared to breastfed infants.

### 3.3. Postbiotics

The definition of postbiotics is derived from the Greek for ‘post’, meaning after, and ‘bios’, meaning life [36]. Different from synbiotics, the microorganisms in postbiotics are no longer alive. Postbiotics are a type of compounds released from food components or microbial constituents by microorganisms and mainly include compounds from bacterial metabolism such as indoles and SCFAs, complex molecules from food compounds, especially enzyme products, and components released from lysed cells including membrane proteins, extracellular vesicles, enzymes, organic acids, lipids, vitamins, peptides, and exopolysaccharides (EPSs) [22,37]. The biological functions of postbiotics have been reported with regard to lipid metabolism, immunity and inflammation effects, antidiabetic properties, and intestinal health [38]. For now, there are already some postbiotics-containing products on the market such as CytoFlora^®^ for immune system support (cell wall fragments of *L. acidophilus*, *Lactiplantibacillus plantarum*, and *L. rhamnosus*), Bactistatin^®^ for intestinal balance (*Bacillus subtilis* VKPM V-2335 metabolites), and Pro-Symbioflor^®^ to improve the digestive system (supernatants and lysates from *Escherichia coli* (DSM 17252) and *Enterococcus faecalis* (DSM 16440)) [38]. The benefits of postbiotics can be used to target specific infant groups such as preterm infants and young children with similar influences as human milk [39].

Several studies reported about the specific fermentation process for delivering postbiotics using two specific types of food-grade lactic acid microorganisms, *B. breve* C50 and *Streptococcus thermophilus* 065, with scGOSs and lcFOSs [40,41]. Many studies have demonstrated the beneficial effects of postbiotics from *B. breve* C50 and *S. thermophilus* O65 on an infant’s growth and intestinal health. For example, a previous study reported that preterm infants fed with fermented formula containing postbiotics from *B. breve* C50 and *S. thermophilus* O65 showed higher levels of *Bifidobacterium* and less risk of abdominal distension and rectal bleeding [42]. These health benefits are thought to occur through the bioactive compounds produced by the microorganisms. In an in vitro study, *B. breve* and *S. thermophilus* postbiotics had anti-inflammatory properties for intestinal cells [43]. They reported that the lipopolysaccharides and other metabolites released from *B. breve* and *S. thermophilus* were resistant to digestive enzymes and to crossing the intestinal barrier, and significantly inhibited TNF-α, LPS-FITC, THP-1, and NF-κB activation. Moreover, this fermented infant formula supplemented with postbiotics increased the sIgA levels in the gastrointestinal tract compared to the formulas with prebiotics or synbiotics, but they were still significantly lower than in breastfed infants [44]. Together, these studies have reported that the health benefits from postbiotics can be explained by the combination of bioactive compounds from *B. breve C50* and *S. thermophilus O65* and prebiotics with *Bifidobacterium*.

Also, some other postbiotic infant formulas have been studied for their in vitro, in vivo, and clinical health benefits. For example, the potential benefits of postbiotics produced by *B. animalis* spp. *lactis* (CECT 8145 BPL1TM) and *L. paracasei* (CBA L74) have been reported. In an in vivo study with baby fecal slurries, this fermented formula has been reported to have beneficial impacts on fat deposition in Caenorhabditis elegans and increases in SCFAs, especially acetate and lactate [45]. Moreover, a randomized, multicenter, double-blind, parallel, and comparative clinical trial was performed to evaluate the safety, efficacy, and tolerability of postbiotics from HT-BPL1. Decreasing risks of atopic dermatitis, bronchitis, and bronchiolitis episodes were observed among infants fed with fermented formula with postbiotics from HT-BPL1 [46]. As for postbiotics produced from *L. paracasei* (CBA L74), fermented formula with this postbiotic showed protective effects against colitis and pathogens (Salmonella) and the inhibition of pro-inflammatory cytokines in favor of anti-inflammatory cytokines [47]. Clinical studies showed the safe use of bovine milk fermented with *L. paracasei* (CBA L74) for infants, and they found increases in concentrations of fecal immune-active peptides and proteins (α-defensin, β-defensin, sIgA, and cathelicidin LL-37), which led to the activation of the innate and acquired immune system in infants [48]. With regard to the gut microbiota and intestinal health, a study reported that the use of a formula fermented with *L. paracasei* (CBA L74) was safe and well tolerated, and a similar microbiota composition to breastfed infants was found in this fermented formula with a reduction in fecal bacterial diversity and the sIgA level and a metabolomic profile [49].

Overall, many studies have demonstrated the safety and tolerability of fermented infant formula with postbiotics from inactivated microorganisms, and these fermented formulas not only benefitted intestinal health and the gut microbiota but also promoted the development of the immune system of infants. Although the microbial inactivation may be performed using thermal or non-thermal technologies, the complex bioactive compositions released from microorganisms are hard to control, thus there are still unknown compounds released from inactivated microorganisms, requiring more experiments to verify their short- and long-term influences on an infant’s health.

### 3.4. Applications of Different Biotics in Infant Formula

The main health benefits of prebiotics, synbiotics, and postbiotics are summarized in Figure 2. Based on the current knowledge, it is hard to provide a comparative analysis of the potential clinical effects of pre-, syn-, and postbiotics on infant health. There are a few studies reporting the slightly higher influences of synbiotics or postbiotics on the gut microbiota, compared to prebiotics [50]. Compared to prebiotics, supplements with synbiotics can promote the colonization of *Bifidobacterium* and lead to compositional differences in gut microbiota [51].

Postbiotics may increase the potency of active microorganisms and turn them into functional ingredients. Rodriguez-herrera et al. (2019) compared the health impacts of breastfeeding with different infant formulas including a 30% fermented infant formula (containing *B. breve C50* and *S. thermophilus 065*), a specific prebiotic mixture (short-chain GOSs and long-chain FOSs (9:1, 0.8 g/L)), and a standard formula [52]. They showed that fermented infant formula resulted in similar outcomes as breastfed infants in their daily weight increase, stool consistency, and the risk of infantile colic. This was also reported by another study, which mentioned that the bioactive compounds from *B. breve C50* and *S. thermophilus O65* inhibited the growth of *Bacteroides distasonis*, *C. lituseburense*, and *C. histolyticum* groups, which are bacteria that may lead to infection, enterotoxin production, and other pathogenic properties [44]. Supplementation with postbiotics seem to have better influences on gut microbiota, the immune system, gut diseases, and allergies in high-risk infants, compared to other biotics.

However, the mechanistic insights, safety profile, regulatory issues, and other key elements have been considered for the application of biotics in infant formula. The safety profiles of biotics in infant formula are still a matter of debate. Early clinical studies suggested that using probiotics and/or symbiotic was safe to infants and the specific evaluated strains, dosages, and duration may have resulted in different outcomes for the safety profiles [53]. For example, recent studies reported that using probiotics may cause systemic infections, the acquisition of antibiotic resistance genes, and interference with gut colonization in neonates [54]. Furthermore, a meta-analysis searched for the use of postbiotics in infant formula from PubMed, Embase, Web of Science, and ProQuest until February 2023 [55]. They reported that supplements of postbiotics in formula increased the concentration of stool SIgA and did not increase the incidence of SAEs, infantile colic, flatulence, diarrhea, vomiting, abdominal pain, and gastrointestinal disorders in infants, thus could be considered as safe for infants. However, the limited number and duration of clinical trials, the postbiotics procedure, the dosage of postbiotics, and other factors may influence the safety profile of postbiotics. This reveals that the regulatory consideration of biotics plays a critical role in infant formula production. However, there are several regulatory issues for the applications of biotics in infant formula such as a lack of standardized regulations across different areas [56]. Moreover, the labeling of biotics is essential for both healthcare professionals and consumers. The labeling regulations of probiotics products should include the genus, species, and strain designation for each probiotic, the effective dose, health claims permitted by law, proper storage conditions, corporate contact details, and other information, while the labels for many probiotics products do not provide enough information on the microbial strains in real life [57]. As for postbiotics, the International Scientific Association for Probiotics and Prebiotics (ISAPP) gave a flexible enough definition of postbiotics, which did not require any particular health benefit, dominant composition, target population, or specific regulatory status [58]. However, recently ISAPP reported the regulatory progress of postbiotics from a global perspective, and they highlighted the discrepancy in postbiotic definitions, which resulted in the regulatory issues in postbiotics products [59]. Accordingly, the main challenges for the regulation of postbiotics in infant formula will be a generally accepted and unitary definition and labels including the ingredients, production procedure, and possible risk [58,60].

In summary, this section described the applications of prebiotics, synbiotics, and postbiotics in infant formula, and their short- and long-term clinical benefits for infants. It seems that supplementation with synbiotics and postbiotics provided better effects on gut microbiota, immune modulation, anti-inflammation, and the risk reduction of diseases, and were especially suited for high-risk infants. An important difference between probiotics and synbiotics is that the bacterial viability in postbiotics is not essential for its health benefits, indicating that the application of postbiotics may be more convenient for foods that cannot carry viable bacteria. Moreover, the comparison regarding the health benefits between synbiotics and postbiotics has hardly been investigated. According to the current clinical data, postbiotics may have more health benefits when supplemented in infant formula or functional foods, because of their variety of bioactive components. However, the safety and regulatory issues have to be considered before the application of biotics in infant formula, especially through large-scale population and long-duration safety clinical trials and stringent regulations for manufacturers.

## 4. Dietary Polysaccharides as Biotics

For now, there are plenty of studies reporting the structures of dietary polysaccharides from natural sources and their biological functions for infants and human adults. However, these dietary polysaccharides are not legally allowed in commercial infant formula for now, which has limited the development of infant formula and applications of multiple biotics. Therefore, this section will combine and summarize the current studies in this area (Table 1), and thereby provide more information about the development of prebiotics, synbiotics, and postbiotics to benefit infants in the future (Figure 3). Especially for postbiotics, the current studies are focused on the changes in fermented microorganisms, and the variations in fermented substrates for postbiotics as prebiotics are hardly mentioned. Therefore, it is important to strictly select the bacterial strain and the variety of substrates for fermentation.

### 4.1. Pectin

Pectin is a component that can be isolated from the primary and secondary cell wall of many vegetables and fruits, using a range of chemical and/or enzymatic methods. Pectin belongs to a heteropolysaccharide with α-1,4-linked galacturonic acid (GalA) residues, and contains different structural characteristics such as homogalacturonan, xylogalacturonan, apiogalacturonan, rhamnogalacturonan I (RG-I), and rhamnogalacturonan II (RG-II). The rhamnose residues in RG-I and RG-II can be branched with different glycosidic linkages, which lead to the complex and variable structures of RG-I and RG-II. These variable and complex pectin structures from vegetables and fruits result in different biological functions for human health. Pectins are considered to greatly impact the health of the gastrointestinal tract and have many health benefits, such as inducing the modulation of the immune system and the prevention of pathogen adhesion.

Normally, these complex plant polysaccharides cannot be fermented by gut microbiota, because these microorganisms prefer to utilize specific components and/or branched glycans. There are a variety of studies in vitro and in vivo reporting that pectin can influence the composition and diversity of the gut microbiota [77,78]. Due to the complex structure of pectin, the influences of pectin on the growth of gut microbiota are different based on its structure (RG-I and RG-II) and degree of methyl esterification (DM) [78]. Generally, populations of beneficial bacteria, such as *Faecalibacterium prausnitzii*, *Coprococcus*, and *Bifidobacterium*, either increased or decreased based on the types of pectin [62]. For example, arabinan, arabinogalactan, and RG-I from pectin significantly increased the growth of *Bifidobacterium* and *Bacteroides* [79]. However, no *Bifidobacterium* species is reported to directly metabolize pectin, and it is possible that other gut commensals such as *Bacteroides* can first digest these large pectin polymers into mono- and oligosaccharides [80]. This reveals that the fermented bacteria for pectin degradation may be different with other substrates (GOSs/FOSs) for postbiotics production. Previous studies have reported that *Bacteroidales*, *Prevotella*, *F. prausnitzii*, *Enterobacteriaceae*, and *Clostridium* can produce a large variety of pectinases to utilize pectin [62]. A previous study reported that pectin-derived acidic oligosaccharides had antioxidant effects, which were utilized in infant formulas to reduce diarrhea, enhance mineral absorption, and facilitate calcium ion uptake [61]. A study showed that pectin from orange peel with a lower DE stimulated the growth of *Bifidobacterium* and inhibited the growth of *C. perfringens* and *B. fragilis* in fecal fermentations [81]. Moreover, several studies show that pectin can increase the abundance of *Lactobacillus*, *Bacteroides*, and *Prevotella*. In contrast, an in vivo study in piglets showed that pectin decreased the relative abundance of the genus *Lactobacillus* and increased that of *Prevotella* in the colon. It is also reported by other studies that pectin can decrease the abundance of *Lactobacillus* [82]. Larsen et al. (2018) reported that bacterial resistance was related to specific interactions between pectin and the glycoproteins on the bacterial cell surface, and these interactions were influenced by the pectin chain conformation and structure [83]. This indicates the importance of pectin structures, the molecular weight, and the DM for bacterial resistance in the gut. These variations in gut microbiota induced by pectin lead to differences in SCFA production. The fermentation of pectin by gut microbiota mainly leads to the production of butyrate, propionate, acetate, lactate, and other metabolites [84], thereby impacting gut health.

Pectin can impact different aspects of the gastrointestinal immune barrier. Similarly to influences on the gut microbiota, the impact of pectin on the gastrointestinal immune barrier and the development of the intestinal immune system are related to the structure of pectin [77]. For example, pectin with a lower DM can penetrate the mucus layer and stimulate goblet cells to produce and secrete mucus [85]. However, pectin with a higher DM can interact with mucins to strengthen the mucus layer and thus protect the epithelium. Furthermore, a few studies reported that specific structural characteristics of pectin have positive effects on the epithelial cell layer integrity, the activation of innate immune cells, and other benefits on immune system development [77]. With regard to other functions, a study demonstrated that pectin increased the recognition memory in mice via upregulated IL-6 and IFN-γ in mouse hippocampi, which led to a decrease in STAT3 and increase in pSTAT3 protein levels [86]. Although these health benefits of pectin are based on the results from in vitro and in vivo studies, many pre-clinical and clinical studies are still required to verify the influences of pectin on infant health specifically.

Because pectin cannot be directly utilized by *Bifidobacterium*, it limits the prebiotic functions of pectin for the gut microbiota, and therefore the fermentation of pectin as a postbiotic in infant formula may be a more useful strategy. This is in line with numerous studies that have indicated the potential applications of pectin as a postbiotic in infant formula production. These studies have demonstrated the applications of pectin in infant formula, although it is not yet commercially available. For example, in 2008, a clinical safety study of pectin in infant formula was reported, which showed that it can reduce the microbiome changes in infant feces caused by the cessation of breastfeeding [65]. Later, in 2021, the panel on Food Additives and Flavourings (FAF) assessed the safety of pectins (E 440i and E 440ii) as additives in food for infants below 16 weeks of age [66]. They reported the safe use of pectin (E 440) for special medical purposes as a supplement in infant formulas. Another study showed that pectin in infant formula could reduce the risk of serious infectious diseases caused by endogenous bacterial infections [63]. There are some pectins from dietary foods which are reported to benefit the growth of beneficial microorganisms in the gut, and these polysaccharides may have a potential application as prebiotics in infant formula production. A study reported that compared to FOSs, RGI-enriched pectin polysaccharides from okra promoted the growth of *Bacteroidetes*, and the relative abundance of *Firmicutes* was decreased [64]. Moreover, several studies demonstrated that pectin can increase the survival of *Lactobacillus* spp. in the gastric solution, indicating that the pectin-and-Lactobacillus-based synbiotics may have potentially interesting applications in aiming to improve the gut microbiota [83].

In summary, the health benefits of pectin, such as on gut microbiota and immune modulation, have been widely reported, with its functions really depending on the structure and source of pectin. Importantly, pectins may potentially be applied as postbiotic production substrates in infant formula in the future, while the selection of pectin and its safety profile still requires more studies.

### 4.2. Inulin-Type Fructans

Inulin-type fructans (ITFs) are a type of prebiotic, and are storage carbohydrates in many plants and fruits such as onion, wheat, and Jerusalem Artichoke. ITFs belong to linear polydisperse carbohydrates linked by β-(2-1) glycosidic linkages, and their industrial production is mainly from chicory (Cichorium intybus) roots [87]. Based on the distribution of chain lengths and the mode of production, ITFs can be classified as inulin and oligofructose. These ITFs refer to a mixture of fructan chains with a degree of polymerization up to more than 60 units.

Various studies have reported the functions of ITFs for infants, especially at a higher dosage, such as for the gut microbiota composition and prevention from infections [67,87,88]. In 2011–2013, ClinicalTrials (ClinicalTrials.gov) evaluated the effects of ITFs in infant formula on gut health for 3–4-month-old infants. They reported that supplementation with inulin promoted the growth of beneficial bacteria, such as *Bifidobacterium*, in stools. For example, a double-blind, randomized, placebo-controlled, and parallel trial demonstrated that the application of infant formula with ITFs and oligofructose increased the level of *Bacteroides*, *Bifidobacterium*, and Enterobacteriaceae to levels similar to breastfeeding [89]. They also reported that supplementation with ITFs resulted in softer and more frequent stools, which may partially be explained by the higher *Bifidobacterium* colonization. However, according to the European Society for Paediatric Gastroenterology, Hepatology and Nutrition, supplementation with prebiotics increased the risk of dehydration in some infants [90]. A study with 164 Indonesian infants showed that if they were fed with 0.4 g inulin/100 mL infant formula, the abundance of *Bifidobacterium* and *Lactobacillus* increased, and the stool pH decreased [91]. Although inulin promoted the growth of most known butyrate producers, such as the *Bifidobacteriaceae*, *Lachnospiraceae*, and *Ruminococcaceae* families, no significant increases in fecal butyrate levels were found with inulin supplementation [68].

The supplementation of infant formula with inulin has been specifically considered for high-risk infants. For example, a synbiotics formula containing ITFs has been shown to improve gastrointestinal health, such as acute gastroenteritis, inflammatory bowel diseases, infantile colic, abdominal pain disorders, constipation, the prevention of infections by *Helicobacter pylori*, and allergies [69]. Moreover, *Lactobacillus* families such as *L. plantarum*, *L. fermentum*, and *L. rhamnosus* can utilize ITFs, inhibiting *E. coli* and other pathogens, thereby showing their potential applications for the development of effective anti-diarrheal synbiotic infant formula [70]. Besides that, inulin was reported to expedite the fermentation of 2′-FL by gut microbiota, and as an immune modulator impacted both dendritic cell and T cell cytokine responses for immune-compromised infants [71]. For preterm infants, there are limited studies reporting the effects of inulin on their health, especially on the prevention of NEC. However, due to the variations in inulin and probiotics strains used in various studies, it is difficult to give a certain impact of inulin on NEC. Two meta-analysis papers suggested that prebiotics including GOSs, FOSs, and inulin may result in little or no difference in the risk of NEC, death, or serious infection for preterm infants, while large and high-quality trials are required to provide more sufficient evidence for this in the future.

### 4.3. β-Glucan and Resistant Starch

The previous sections have mentioned the applications of pectin and inulin in infant formula, while there are many other non-digestible carbohydrates that may be beneficial to the gut microbiota and immune system of infants. For example, β-glucan is a soluble dietary fiber from oats, barley, bacteria, yeast, algae, and mushrooms [92]. In oats and barley, β-glucans are polymers of D-glucose linked via β-(1→4) and β-(1→3) glycosidic bonds, and D-glucose is linked via β-(1→6) and β-(1→3) glycosidic bonds in yeast and mushrooms. Of these, β-glucans linked with β-(1→3) are almost resistant to the conditions in the human gastrointestinal tract and can therefore reach the small intestine (duodenum). Various studies have demonstrated the resulting prebiotic functions of β-glucan in humans, also showing that the molecular weight of β-glucan may be the rate-limiting factor impacting its clinical functions [75]. As for the gut microbiota composition, numerous in vitro and in vivo studies demonstrated that different sources of β-glucan had slightly different influences on gut microbiota, and mostly contributed to the growth of *Bifidobacterium* and *Lactobacillus* [93]. For example, barley β-glucan increased the abundance of *Bifidobacterium* and *Bacteroides*, resulting in a higher concentration of total SCFAs, especially acetate and butyrate [74]. This is similar to a study reporting the functions of mushroom β-glucans on gut microbiota, which showed increased populations of *Bifidobacterium* and *F. prausnitzii* as well as levels of SCFAs [94]. Furthermore, an in vivo study showed that β-glucan from *Trametes versicolor* increased the levels of gut microbes, such as *Actinobacteria*, *Lactobacillus johnsonii*, and *Lactobacillus murinus*, and reduced the proportion of *Klebsiella oxytoca* and *Klebsiella* in an NEC model [76]. They reported that β-glucan ameliorated the intestinal injury of NEC mice by the inhibition of the expression of TLR4, NF-κB, IL-1β, IL-6, and TNF-α and increasing the expression of IL-10, ZO-1, Occludin, and Claudin-1. Recently, a few studies reported the potential beneficial effects of β-glucan on infant health. For example, an in vitro fermentation study with oat β-glucan with the infant fecal inoculum of 2- and 8-week-old infants showed that oat β-glucan increased the abundance of Enterococcus and modulated the immune cells, reducing intestinal inflammation [73]. It is interesting to note that the immune modulating functions of β-glucan were partly similar to those of HMOs. This is mainly because β-glucan can be trapped by different pattern recognition receptors such as dectin-1 and toll-like receptors (TLRs) of macrophages and dendritic cells, generating numerous cytokines and activating immune cells to stimulate specific immunity through T and B cells [73,95]. Akkerman et al. (2020) reported that fermented oat β-glucans with a smaller molecular weight were better utilized by gut microbes, such as *Enterococcus* and *Lactobacillus*, and supported the development of the immune system.

Resistant starch (RS) is mainly defined as a type of starch with α-(1→4) and α-(1→6) glucans that cannot be digested by amylases in the small intestine [72]. The functions of RS are mostly considered as prebiotic, although RS can be utilized by all colonic bacteria non-selectively, and not just by beneficial microbes [96]. A previous study mentioned the benefits on gut microbiota and total SCFA concentrations not only from FOSs and β-glucan, but also from RS [74]. They suggested that RS can be widely fermented from the cecum to the distal colon, thus generating higher concentrations of SCFAs. This is in line with an in vitro fermentation study of the effects of RS on infant gut microbiota, which demonstrated that compared to FOSs, RS increased the concentrations of propionate and butyrate, although resulted in significantly lower levels of acetate [97]. They reported that RS increased the relative abundance of *Bacteroides* and *Bifidobacterium* and decreased the proportions of Proteobacteria and Enterobacter [72]. A quantitative PCR showed that RS increased the total bacteria count and stimulated the growth of *Bifidobacterium*. The effects of RS on infant gut microbiota were different between pre-weaning and post-weaning infants. However, an in vitro study reported that the digestion ability for RS in the gastrointestinal tract was different due to the present of digestible starch fractions and variations in specialized microbes to degrade RS [98]. This may lead to differences in the prebiotic property of RS.

Overall, these studies indicate the prebiotic and especially the immune-modulating functions of β-glucan and RS. The functions of β-glucan reveal the potential applications of it as a postbiotics substrate in infant formula production, although this has not been studied yet. In addition, RS shows potential benefits for the growth of gut microbiota and producing propionate and butyrate, especially for weaning infants.

### 4.4. Other Dietary Fibers

There are still some other polysaccharides reported to have prebiotics functions being used in in vitro and in vivo studies, although they are not yet used in infant formula studies. For example, hemicellulose, as a major constituent in the plant cell wall, is a highly branched low-molecular-weight biopolymer and is mainly composed of pentoses (xylose, arabinose), hexoses (galactose, mannose, glucose), and sugar acids (glucuronic, galacturonic, cinnamic, and methyl galacturonic acid). The two major plant hemicelluloses are xylan and mannan. Several studies reported the prebiotic functions and selective advantage to gut microbiota of oligosaccharides being derived/hydrolyzed from cellulose and hemicellulose, including cello-oligosacchrides (COSs), xylo-oligosaccharides (XOSs), manno-oligosaccharides (MOSs), and arabinoxylo-oligosaccharides (AXOSs). The industrial production of both XOSs and AXOSs is by hydrolysis from the insoluble arabinoxylan fraction. Of these, XOSs are a mixture of oligosaccharides including xylobiose, xylotriose, and xylotetrose containing xylose residues linked by α1–4 bonds, which are normally present in vegetables, fruits, and bamboo shoots [99]. A previous study demonstrated that XOSs had prebiotics effects on the growth of *Bifidobacterium* and *Peptostreptococcus*, and showed beneficial influences on intestinal health and had an anti-inflammatory capacity when used as a mixture of XOSs and GOSs [100]. An in vivo study in rats demonstrated that XOSs increased the relative abundance of beneficial bacteria such as Bacteroidetes and reduced the inflammation induced by a high-fat diet [101]. AXOS is the major non-cellulose polysaccharide in cereals and plants. A previous study demonstrated that AXOSs stimulated the growth of *Bifidobacterium* via the binding protein BlAXBP and ATP-binding cassette (ABC) transporter, and showed a preference for tri- and tetrasaccharides over the xylose monosaccharide [102]. This is similar to a study that reported the prebiotic functions of AXOSs and XOSs for *Lactobacillus* and *Bifidobacterium* being better than those of xylose [103]. The prebiotic functions of AXOSs may be caused by the release of arabinose units by *Bifidobacterium*, without degrading the xylan backbone [99]. Van Den Abbeele et al. (2018) reported that AXOSs increased the abundance of *Bifidobacterium*, *Actinobacteriam*, *Bactteroidetes*, and of propionate [104]. An MOS from yeast mainly contains repeating units of mannose linked with β-1,4-glycosidic bonds and is easily degraded by acid hydrolysis and digestive enzymes in the intestine into α-MOS. An MOS from plants contains mannose and glucose linked with β-1,4-glycosidic bonds as the main chain with galactose as the side branch, linked with β-1,6-glycosidic bonds, which can be hydrolyzed into a different DP of β-MOS [105]. Both types of MOS show beneficial impacts on *Lactobacillus* and *Bifidobacterium* [105,106]. In addition, MOSs can inhibit the growth of pathogens such as *Shigella dysenteriae* [106] and *Clostridium* [107], reducing intestinal infections. COSs are novel functional oligosaccharides with β1-4 glycosidic linkages, which are produced by the chemical and enzymatic hydrolysis of cellulose. An in vivo study with weaned piglets showed that COSs not only increased the level of *Lactobacillus* and decreased the level of Clostridium, but also led to a positive effect on intestinal microflora, mucosal architecture, and nutrient transport [108]. Furthermore, there are several dietary polysaccharides from special mushrooms and plants being reported to have prebiotic functions on the gut microbiota. Mushrooms are a kind of large fungi and contain different sugar compositions and structures to plants. For example, β-glucan from mushrooms can increase the number of beneficial bacteria such as *Bifidobacterium* and *Lactobacillus* [109]. With regard to heterosaccharides, *H. erinaceus* polysaccharides significantly promoted the growth of *Bifidobacterium*, *Faecalibacterium*, *Blautia*, *Butyricicoccus*, and *Lactobacillus*, and inhibited the levels of pathogens such as *Escherichia-Shigella* and *Enterobacter* [110]. Moreover, chitin is a natural component of most cell walls for fungi and yeast, and is mainly composed of glucosamine with β1-4 glycosidic linkages. Several studies reported chitin as being beneficial for human gut health, while its application in infant formula still requires further studies. For example, a double-blind, randomized, cross-over, exploratory study conducted over two 3-week phases showed that chitin–glucan improved postprandial glucose and lipid metabolism and regulated the gut microbiota and SCFA production by decreasing the abundance of Actinobacteria and SCFAs, such as butyric acid [111]. Arabic gum is isolated from the hardened exudates of the plants Acacia Senegal and Acacia seyal, being composed of L-rhamnose, L-arabinose, D-galactose, and D-glucuronic acid, connected by several types of glycosidic linkages. Arabic gum is mainly produced from trees in the Sahel region of Africa. A comprehensive evaluation of Arabic gum as a food additive in infant formulas suggested that there is no safety concern for Arabic gum for the general population and it can be slightly fermented by gut microbiota [112]. Several studies reported the prebiotic functions of Arabic gum. For example, an in vitro fermentation study demonstrated that acacia gum with an average DP of five stimulated the growth of *Bifidobacterium*, *Lactobacillus*, and *Bacteroides* and inhibited Clostridium-induced dysbiosis [113]. Moreover, infants fed with an anti-regurgitation (AR) formula containing locust bean gum showed no looser or more watery stools than the control group [114].

Overall, this section summarized the prebiotics functions of soluble and insoluble dietary fibers for gut microbiota through in vitro and in vivo studies. Various studies have focused on the soluble dietary fiber, especially pectin and inulin, and demonstrated their application in infant formula. A few studies reported the prebiotic effects of oligosaccharides derived from cellulose and hemicellulose, which showed certain properties promoting the growth of beneficial probiotics such as *Bifidobacterium* and *Lactobacillus*, although these results still require more in vivo and clinical studies. These dietary polysaccharides and some special polysaccharides from mushrooms and plants may be interesting sources of substrates for postbiotics production in the future.

## 5. Development of Delivery of Multiple Biotics

Although the previous sections described the functions for many prebiotics in the colon, the lower stability of synbiotics and postbiotics during their passage through the gastrointestinal tract may limit their function. One of the most important reasons is the acidic environment and enzymatic hydrolysis of biotics in the gastrointestinal tract, although the potential influences of these conditions on multiple biotics still remains unknown. This may lead to variations in prebiotic (e.g., GOSs/FOSs) and synbiotic functions and the level and activity of bioactive compounds in postbiotics. However, the applications of encapsulation technology for postbiotics have been hardly studied in infant formula, although it is important to protect the bioactive compounds from postbiotics. The main challenge for postbiotics delivery is the instability of encapsulation technology caused by the smaller molecular size of postbiotics. Therefore, more studies are needed that focus on the specific delivery systems, such as microcapsules and nanoparticles (Figure 4), for multiple biotics to have functions for infants in their colon, especially for their gut microbiota. This section will summarize the recent studies about these delivery systems and discuss the applications of these systems to multiple biotics in infant formula.

### 5.1. Microcapsules

Microcapsule delivery systems have been applied in infant formula for nutrients to avoid their oxidative degradation [115]. Natural or synthetic polymer materials can be used to form the microcapsules in the micrometer to millimeter size range. There are a variety of technologies to form microcapsules, such as spray drying, lyophilization, supercritical fluid precipitation, coacervation, liposomes, ionic gelation, interfacial polymerization, and molecular inclusion complexation [115]. Microcapsule technology has been widely used for probiotics delivery. Encapsulated probiotics, generally including *Bifidobacterium*, *Lactobacillus*, and *Enterococcus*, have been reported in dairy products [116]. For example, Alves et al. (2023) reported the potential applications of *L. reuteri* in skim milk and infant formula as a microencapsulation material using spray drying, protecting (>90%) the probiotics in the infant formula [117]. Moreover, many studies focused on the co-encapsulation of prebiotics with probiotics to form synbiotics in infant formula. Many encapsulation materials with non-digestible polysaccharides including maltodextrin [118], pullulan [119], arabinoxylan [120], pectin [121], and inulin [122] have been reported. Co-encapsulation with RS and alginate had better protective effects and a higher viability for *Lactobacillus plantarum* and *Bifidobacterium lactis* compared to inulin, and this is mainly due to the interactions between RS and Ca-alginate networks [122]. Furthermore, microencapsulation with pectin and sodium alginate showed better protective effects on *B.animalis BB-12* compared to *L. acidophilus LA-5* in gastrointestinal conditions [121]. Moreover, the addition of chitosan could significantly reduce the release of probiotics and prebiotics during the processing of encapsulation and in gastrointestinal conditions. Together, these studies indicated that the choice of encapsulation material can not only prevent the inactivation of probiotics but also protect the prebiotics functions. It is important to note that polysaccharides such as starch are used as encapsulation matrices for the above-mentioned probiotics microcapsules. The polysaccharide-based microcapsules should not only have consistent transit through the gastrointestinal tract [123], but also lead to the precise release of encapsulated nutrients from the microcapsules in the colon by gut microbiota. Therefore, there is an increasing interest in the development of novel encapsulation matrices to generate microcapsules for the delivery of nutrients in infant formula.

As for postbiotics delivery, although the delivery of postbiotics is hardly studied, the dominant components of postbiotics including EPSs and indoles have been reported recently. EPSs from different bacterial strains have been widely used as encapsulation materials for microcapsules. For example, an EPS from *Limosilactobacillus reuteri* in human milk has been used to encapsulate caffeic acid and increase the stability of it [124]. Moreover, an EPS from *L. plantarum* with sodium alginate and calcium chloride dihydrate as encapsulation materials showed better prebiotics effects than FOSs and increased the survival rate of *Lactobacillus acidophilus* [125]. Other macromolecules such as membrane proteins, peptides, and peptidoglycan may contribute to the formation of microcapsules with encapsulation materials. Besides that, Yang et al. (2022) demonstrated the encapsulation of indole-3-propionic acid as a postbiotic with sodium alginate, resistant starch, and chitosan, and they reported the synergy between prebiotics and postbiotics in gut microbiota [126]. Besides that, Arabic gum presents a potential application as an encapsulation material with whey protein, RS, pectin, chitosan, and other polymers for synbiotics and postbiotics in infant formula. This is mainly due to great performance in smaller molecule delivery, such as for lemon oil, anthocyanins, and carotenoids [127].

Spray drying is the most common and economical method used for infant formula production, while a high temperature, dehydration, osmotic pressure, and other treatments may result in low stability and viability for the microcapsules and probiotics [128]. Therefore, the stability of microcapsules for multiple biotics delivery is still the main challenge limiting their applications. The encapsulation materials play critical roles in the stability of microcapsules and the encapsulated bioactive compounds. The co-encapsulation of prebiotics and probiotics has been reported to increase the stability and viability of probiotics during spray drying [116]. As an example, the co-encapsulation of RS, chitosan-coated alginate, and *Lactobacillus plantarum* showed better stability during spray drying and in gastrointestinal conditions [129]. Compared to other combinations of polysaccharides and prebiotics for co-encapsulation, their results showed that RS–alginate-based microcapsules were the most stable and had the highest survival rate (79%) for *Lactobacillus casei* [130]. Moreover, inulin with other materials such as maltodextrin and Arabic gum as co-encapsulation materials has been reported to have better protective effects for probiotics from high temperatures and a long shelf life [131]. The addition of pectin to encapsulation materials had a high thermal resistance (>59%) and rate of survival in acidic conditions (>80%), and resulted in a better release of probiotics at the site of action compared to gelatin [132]. Low DM pectin especially resulted in a more compact and smoother surface for microcapsules and had the highest releasing rate (>95%) of *Lactiplantibacillus plantarum* at pH 7.4 in the human gastrointestinal tract [133]. The stability of prebiotics-based microcapsules for postbiotics delivery has been improved, especially for the small molecular compounds, while the influences from high heating and pressure during spray drying still remain unknown. As an example, the microcapsules composed of sodium alginate, RS, and chitosan provided excellent protection of IPA [126]. Overall, the co-encapsulation of prebiotics with probiotics (as synbiotics) or postbiotics not only increased the health benefits of probiotics and postbiotics on gut microbiota, but also improved the stability of the microcapsules. However, due to the complex constitutions of postbiotics, the detailed applications of microcapsules in postbiotics delivery and the interactions between encapsulation materials and postbiotics still require more studies in the future.

### 5.2. Nanoparticles

During the last decade, nanotechnology has been used to develop and characterize nanoparticles or nano-encapsulation for multiple biotics delivery with a size range of 1–100 nm. Different from other drug carriers, nanoparticles have high stability, a high carrier capacity, high biocompatibility, high bioavailability, and other benefits. The normally used nanotechnologies are divided into top-down, like emulsification, and bottom-up, such as self-assembly, supercritical fluids, and nanoprecipitation [134]. The commonly used nanoparticle materials include distinctive organic compounds such as carbohydrates, proteins, and fats, or inorganics such as oxides of silver, calcium, zinc, magnesium, etc. These inorganics in nanoparticles could be considered for the dietary supplementation of minerals for infants. For example, the nanoparticles with whey protein isolate and calcium chloride showed higher stability and viability (87.6%) for *Lactobacillus* rhamnosus and kept a constant viability after 28 d of storage [135]. The nanocellulose, chitosan, gum Arabic, pectin, and starch have been reviewed to use for nanoparticle production, and using these prebiotics as nano-encapsulation materials also improved the heating stability [136]. The applications of nanoparticles with pectin for different bioactive compounds have been reported, such as for DHA, anthocyanin, vitamins, polyphenols, lactoferrin, and they can be combined with proteins and lipids to improve their stability and controlled release rate [134]. Different from others, chitosan-based nanoparticles showed stronger binding with a layer of mucus because of the interactions between the negatively charged mucin and positively charged chitosan [137]. Although nanoparticle delivery for postbiotics is hardly studied, nano-encapsulation with different prebiotics seems to have great performance in small molecular or bioactive compounds for probiotics [138]. Therefore, the applications of nanotechnology could be considered as a novel delivery strategy for postbiotics to form nanoparticles and combine them with microcapsules in infant formula production.

In summary, this section reviewed the applications of microcapsules and nanoparticles in the delivery of multiple biotics. Spray drying is a fast and economical technology for the production of microcapsules, while high temperature and pressure are the main challenges for the stability and viability of microcapsules. Numerous studies have reported that the co-encapsulation of prebiotics with probiotics or postbiotics can increase the stability of microcapsules, including heat and acidic resistance. Importantly, the addition of prebiotics improved the target release of probiotics or postbiotics in the gastrointestinal tract, which combined with the prebiotics to benefit the gut microbiota. Moreover, the application of nanoparticles may result in high stability, better protective effects, and the high bioavailability of postbiotics in gut microbiota. Other delivery systems such as bilosomes, liposomes, hydrogel, and other nanotechnologies have been widely used for drug delivery, which could be considered as novel targeted delivery/release strategies for multiple biotics delivery in the future. However, the delivery strategy of multiple biotics is still a challenge for the future in terms of being safe, efficacious, robust, and commercially viable.

## 6. Conclusions and Perspectives

To summarize, infant formula plays a critical role in infant gut microbiota development, as well as other short- and long-term health benefits for infants. Over the last few decades, many studies and a few commercial products have demonstrated the use of prebiotics, synbiotics, and postbiotics in infant formula and their influences on gut microbiota. However, the aim of infant formula is to better mimic the health impact of breastfeeding. Recently, several studies reported the structures and prebiotic functions of dietary polysaccharides. Accordingly, the primary purpose of this manuscript was to systematically review the literature regarding dietary polysaccharides and their impacts as prebiotics, synbiotics, and postbiotics in infant formula on gut microbiota. Generally, different biotics presented different effects on gut microbiota. Synbiotics in infant formula better promoted the colonization of probiotics such as *Bifidobacterium* and *Lactobacillus*, while the safety and regulation issues are the dominant challenges for related products. Moreover, due to the presence of other bioactive compounds in postbiotics, they seems to have a better impact on gut health and anti-inflammation for infants, especially those at high risk of diseases. As for dietary polysaccharides, pectin, β-glucans, and oligosaccharides derived from hemicellulose presented similar and even better influences on gut microbiota and SCFA production than the currently used FOS/GOS mixture. However, these results were inconsistent because of the variations in the structures and sugar compositions of these polysaccharides. Furthermore, the practical challenges such as instability and variations in biotics during infant formula production and safety and regulatory issues have to be considered before the application of biotics in infant formula. Moreover, the co-encapsulation of prebiotics with probiotics or postbiotics improved the stability of microcapsules during spray drying, and thus has a good prospect for further development.

Future studies should focus on the applications of different combinations of synbiotics and postbiotics in infant formulas targeting infants at different stages, especially regarding the disruptions to their microbiome in early life such as antibiotic exposure. An interesting development could be postbiotics fermented by different probiotics using dietary polysaccharides as substrates. This would require much more in vitro, in vivo, and clinical studies to evaluate their safety and health benefits for infants compared to traditional formula. Furthermore, advanced colonic delivery strategies for postbiotics should be studied as well.

## Figures and Tables

**Figure 1 nutrients-16-04122-f001:**
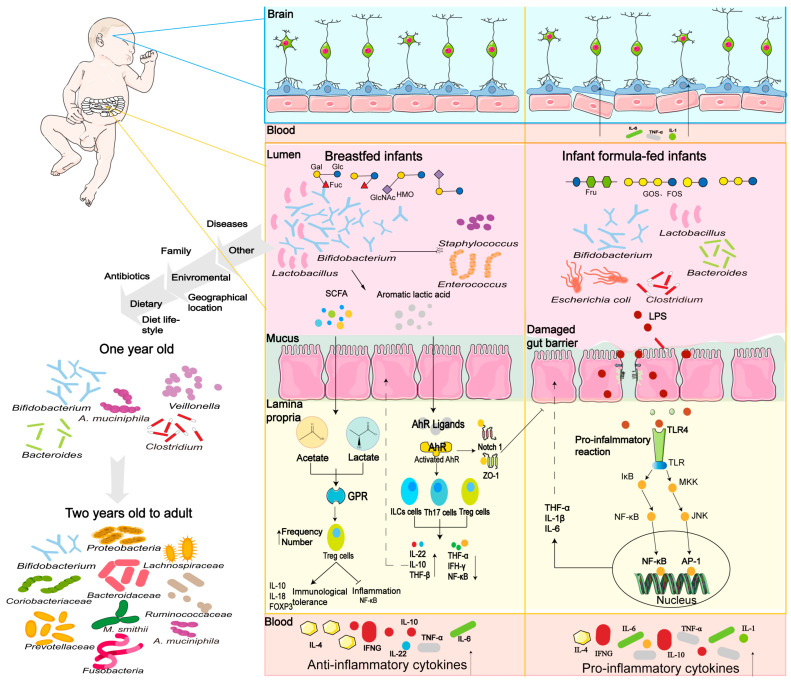
The overview of gut microbiota between breastfed and infant formula-fed infants, and their consequences on gut health and brain development. The meanings for the symbols in this figure : up-regulated (↑), down regulated (↓), galactose (Gal), glucose (Glu), fucose (Fuc), N-acetylglucosamine (GlcNAc), fructose (Fru).

**Figure 2 nutrients-16-04122-f002:**
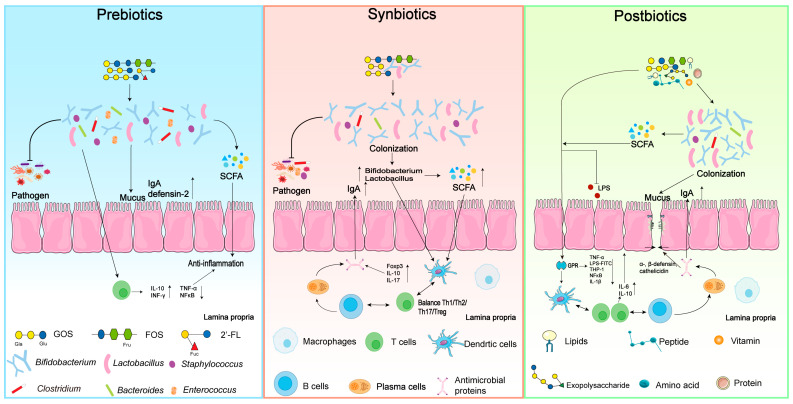
The dominant health benefits of prebiotics, synbiotics, and postbiotics on infant gut microbiota, pathogen infection, inflammation, and the gastrointestinal immune barrier. The meanings for the symbols in this figure: up-regulated (↑), down regulated (↓), galactose (Gal), glucose (Glu), fucose (Fuc), N-acetylglucosamine (GlcNAc), fructose (Fru).

**Figure 3 nutrients-16-04122-f003:**
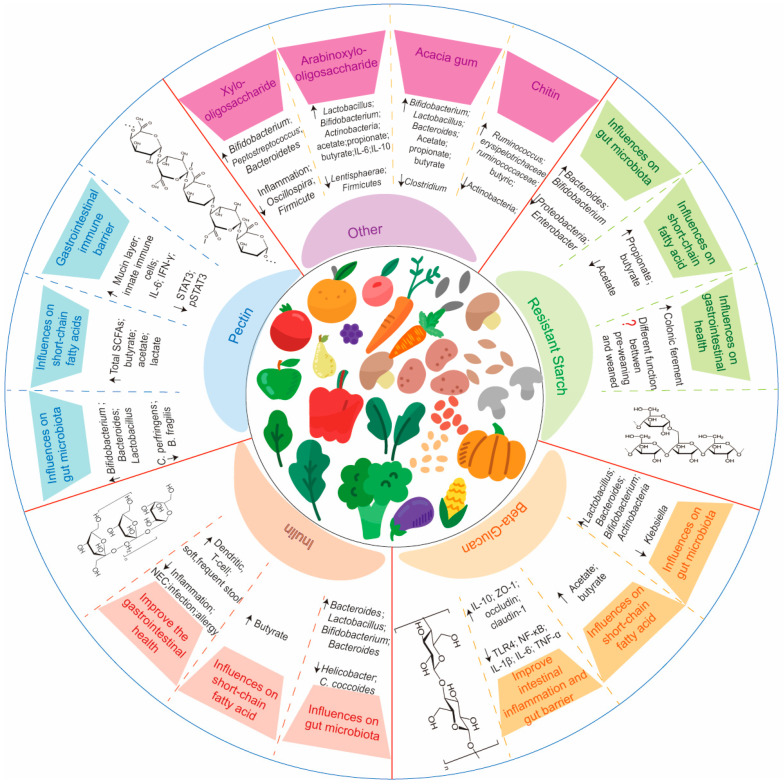
The structures and biological functions of the summarized polysaccharides. The meanings for the symbols in this figure : up-regulated (↑), down-regulated (↓), unknown (?).

**Figure 4 nutrients-16-04122-f004:**
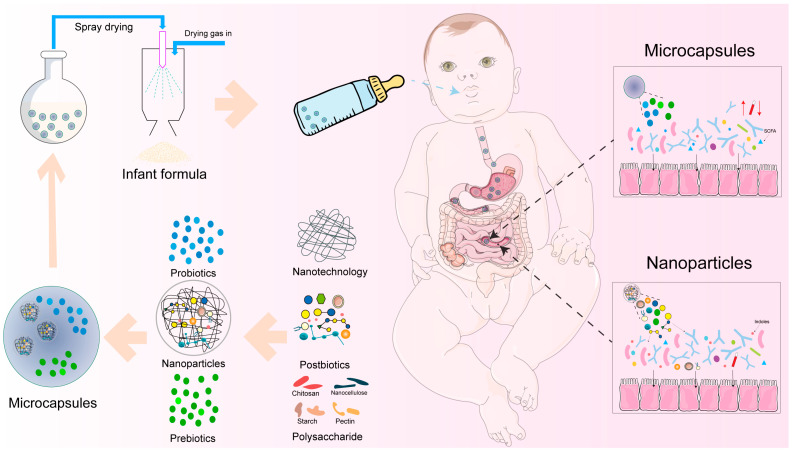
The delivery and release of multiple biotics with microcapsules and nanoparticles in the infant formula.

**Table 1 nutrients-16-04122-t001:** Effects of different types, origins, and structures of dietary polysaccharides on gut health. The meanings for the symbols in this table: up-regulated (↑), down-regulated (↓), not applicable (N/A).

Substrates	Origin	Molecular Structure/Size	Other Parameters Described in the Studies	Study Groups	Health Outcomes	Reference
Pectin	*Okra (Abelmoschus esculentus*)	DW: 37.05%	Human gut microbiota that can produce pectin lyase to degrade high-methoxyl pectin; RG I can be degraded by endo-PLs (endo-polysaccharide lyases) encoded by RGI-PUL in *Bacteroides*	In vitro study with humans	Gut microbiota: *Bacteroidetes* ↑, *Firmicutes* ↓; compared to FOS group, butyric acid and total SCFAc ↑	[60]
Inulin	*Cichorium intybus*	Mixture	Meta-analysis of nine original articles	Human adults >18 years old	Effect of inulin on gut microbiota: *Bifidobacterium*, *Anaerostipes*, *Faecalibacterium*, *Lactobacillus ↑*, *Bacteroides* ↓	[61]
Inulin	Industrial production	Mixture of 50% oligofructose (DP < 10) and 50% long-chain inulin (DP > 10)	Same amounts of lactose, protein, fat, and micronutrients	Infants exclusively breastfed during at least the first 4 months of their life; control: 67 male and 57 female; inulin group: 62 male and 66 female; breastfed: 66 male and 70 female	For inulin-fed infants: total microorganism counts ↓; hardly detected *C. coccoides* and *C. leptum*; *Bacteroides*, *Bifidobacterium*, *Enterobacteriaceae* ↑; softer and more frequent stools; urea concentration ↓	[62]
β-glucan	*Trametes versicolor*	β-(1,4) main chain and β-(1,3) side chain, with β-(1,6) side chains	N/A	In mice: control, NEC induction, and drug treatment	Gut microbiota: *Actinobacteria*, *Clostridium butyricum*, *Lactobacillus johnsonii*, *Lactobacillus murinus*, and *Lachnospiraceae bacterium mt14 ↑*, *Klebsiella oxytoca*, and *Klebsiella ↓*; *protected against NEC: inhibited intestinal inflammation of TLR4*, *NF-κB*, *IL-1β*, *IL-6*, and *TNF-α; increased IL-10*, *ZO-1*, *Occludin*, and *Claudin-1*	[63]
β-glucan	Oat	Commercial enzyme preparation with endo-1,3(4)-β-glucanase, resulting in oligomers with DP ranging from 2 to 5	Being most abundant with glucosyl-(1→3)-β-d-cellotriose	In vitro fermentation with infant fecal inoculum of 2- and 8-week-old infants, evaluated with dendritic cells and CD34+ progenitor cells	Gut microbiota: *Enterococcus*, *Escherichia-Shigella,* and *Clostridium Sensu Stricto I* ↑, Bacillus ↓; SCFAs ↑; immune receptor dectin-1, IL-10 ↑; MIP1α/CCL3, IL-1β, IL-6, TNFα ↓	[64]
β-glucan	*Barley*	Low molecular weight, MW 12 kDa	Two barley lines are BM and BG	Four-week-old rats	Gut microbiota: *Bifidobacterium*, *Bacteroides* ↑, *Firmicutes* ↓; Total SCFAs, acetate, n-butyrate ↑	[65]
β-glucan	Mushroom	Different contents in total and β-glucans	29 strains of *Pleurotus ostreatus*, *P. eryngii*, *P. nebrodensis*, *P. citrinopileatus*, *Hericium erinaceus,* and *Cyclocybe cylindracea*	In vitro fermentation with human fecal microbiota	Gut microbiota: *Bifidobacterium*, *F. prausnitzii* ↑; SCFAs: acetate, propionate, and butyrate ↑; aromatic amino acids, phenylalanine, tyrosine, gamma-aminobutyric acid ↑	[66]
Resistant starch	*Maize*	70% amylose and 30% amylopectin, 50 g resistant starch/100 g	N/A	In vitro fermentation with infant fecal inocula, pre-weaning and weaned	Pre-weaning: alpha and beta diversity, no differences, *Enterococcus*, *Bifidobacterium*, total bacteria, total SCFAs, acetate ↑; weaned: alpha and beta diversity, *Bacteroides*, *Bifidobacterium*, total SCFAs, acetate, propionate, butyrate ↑, *Proteobacteria*, *Enterobacter* ↓	[67]
Xylo-oligosaccharides	N/A	N/A	N/A	Rats	Gut microbiota: *Bacteroidetes*, *Preovtella*, *Proteobacteria* ↑, *Oscillospira*, *Firmicutes* ↓	[68]
Arabinoxylo-oligosaccharide	Wheat bran	Average DP of 3–5	Hydrolyzed by xylanases	In vitro fermentation	*Lactobacillus*, *Bifidobacterium*, lactic acid, acetic acid ↑	[69,70]
Arabinoxylo-oligosaccharide and inulin	Sigma	Inulin: 3 and 60 with an average of 10; an AXOS-rich extract: 66% AXOS, an average degree of polymerization of 6	N/A	In vitro fecal fermentation	Gut microbiota: *Bifidobacterium*, *Actinobacteria*, *Bactteroidetes* ↑, *Lentisphaerae*, *Firmicutes*, *Verrucomicrobia*, *Proteobacteria* ↓; SCFAs: acetate, propionate, and butyrate levels ↑; positively affected immune function: IL-6, IL-10 ↑	[71]
Acacia gum	Trunks and branches of *Acacia senegal* (*A. senegal*) and *Acacia seyal* (*A. seyal*) trees	Average DP 5	A. several contains higher arabinose and 4-O-methyl glucuronic acid	In vitro fecal fermentation	Gut microbiota: *Bifidobacterium*, *Lactobacillus*, *Bacteroides* ↑, *Clostridium* ↓; SCFAs: acetate, propionate, and butyrate ↑	[72]
Chitin–glucan	Fungi	A branched β-1, 3/1, 6 glucan that is linked to chitin via β-1, 4 linkages	N/A	A double-blind, randomized, cross-over, exploratory study conducted over two 3-week phases with 15 subjects	Gut microbiota: *Ruminococcus*, *Erysipelotrichaceae*, *Ruminococcaceae*, *Eubacterium ventriosum* ↑, *Actinobacteria* ↓; SCFAs: butyric acid ↑; postprandial glucose and lipid metabolism ↑	[73]
Cello-oligosacchrides	Cellulose from wheat straw	β-1,4-glucoside bond	N/A	144 weaned piglets	Gut microbiota: *Lactobacillus* ↑, *Clostridium* ↓; intestinal morphology: villus height, crypt depth, villus surface area ↑	[74]
Manno-oligosaccharide	Copra meal	β-MOS	Hydrolysis by β-mannanase from Bacillus circulans	In vitro fecal fermentation	Gut microbiota: *Bifidobacterium*, *Lactobacillus* ↑, *Shigella dysenteriae* ↓	[75]
Polysaccharides	*Hericium erinaceus*	Fructose: mannose: glucose: galactose: HEP-30 is 0.3:1.3:9.8:0.3; HEP-50 is 1.7:0.5:10.6:10.4; HEP-70 is 1.2:1.3:23.7:0.3	Extracted from hot water-soluble with 30%, 50%, and 70% ethanol concentrations (*v*/*v*)	In vitro fecal fermentation	Gut microbiota: *Bifidobacterium*, *Faecalibacterium*, *Blautia*, *Butyricicoccus* and *Lactobacillus* ↑, *Escherichia-Shigella*, *Klebsiella and Enterobacter* ↓; Total SCFAs, acetic acid, propionic acid, valeric acid ↑	[76]

## Data Availability

No new data were created or analyzed in this study. Data sharing is not applicable to this article.

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
