# Peer review of "A Comprehensive Review on Dietary Polysaccharides as Prebiotics, Synbiotics, and Postbiotics in Infant Formula and Their Influences on Gut Microbiota"

_nutrients, 2024, doi:10.3390/nu16234122_

Round 1

Reviewer 1 Report

Comments and Suggestions for Authors

General comments:

1. The scope and objectives of the review should be clearly defined and carefully followed to organize the review contents.

2. Technical terms should be carefully and correctly used or stated.

3. Major views and conclusions should be drawn based on accurate and reliable information sources.  

4. For a rapidly growing topic, the references should be updated till 2024, and relevant contents should be reviewed.

Specific comments

1. The topic on benefits and applications of dietary polysaccharides in infant formulas is interesting and important. However, there are some major discrepancies and misleading points in the review contents with what the title refers.

- Few of the polysaccharides reviewed in this paper were actually used in infant formula or tested on infants. Instead, most were tested on in vitro models or on adults.

- Most of the prebiotics revied in section 3.1 belong to oligosaccharides, rather than polysaccharides.

- Dietary polysaccharides cannot be classified as postbiotics by definition.

2. The safety and regulatory issues for supplementation of polysaccharides and postbiotics to infant formulas are very important, and should also be covered in a comprehensive review.

3. “Over the last few decades, many studies and commercial products have demonstrated the use of prebiotics, synbiotics, postbiotics in infant formula and their influences on gut microbiota.” Check carefully to verify if this this true if “many commercial products have demonstrated. To my knowledge, only a few prebiotics have been approved to use in commercial infant formulas.

Author Response

General comments:

  1. The scope and objectives of the review should be clearly defined and carefully followed to organize the review contents.

Author response: Thanks for your suggestion. We have revised the scope and objective of this paper and organized the following contents.

  1. Technical terms should be carefully and correctly used or stated.

Author response: Done as suggested. Thanks.

  1. Major views and conclusions should be drawn based on accurate and reliable information sources.  

       Author response: Thanks for your comments. We have revised the main conclusions to be more critical.

  1. For a rapidly growing topic, the references should be updated till 2024, and relevant contents should be reviewed.

       Author response: Thanks for your suggestions. We updated our references until 2024。

Specific comments

  1. The topic on benefits and applications of dietary polysaccharides in infant formulas is interesting and important. However, there are some major discrepancies and misleading points in the review contents with what the title refers.

- Few of the polysaccharides reviewed in this paper were actually used in infant formula or tested on infants. Instead, most were tested on in vitro models or on adults.

- Most of the prebiotics revied in section 3.1 belong to oligosaccharides, rather than polysaccharides.

- Dietary polysaccharides cannot be classified as postbiotics by definition.

 Author response: Thanks for your suggestions. Actually, all the prebiotics used in infant formula are oligosaccharide because they are earlier to be digested. Therefore, in this case, the polysaccharide was considered as the precursor for prebiotics/synbiotics/postbiotics in infant formula. We also made these information more precise in the text.

  1. The safety and regulatory issues for supplementation of polysaccharides and postbiotics to infant formulas are very important, and should also be covered in a comprehensive review.

  Author response: Thanks for your comments. We have added the safety and regulatory issues for biotics products in section 3.4.

  1. “Over the last few decades, many studies and commercial products have demonstrated the use of prebiotics, synbiotics, postbiotics in infant formula and their influences on gut microbiota.” Check carefully to verify if this this true if “many commercial products have demonstrated. To my knowledge, only a few prebiotics have been approved to use in commercial infant formulas.

 Author response: Thanks for your suggestion. This is kind of misunderstanding. It means many studies and a few commercial products about synbiotics and postbiotics in infant formula. We have revised this sentence to be more precise.

Reviewer 2 Report

Comments and Suggestions for Authors

The review titled: “A comprehensive review on dietary polysaccharides: As prebiotics, synbiotics, and postbiotics in infant formula and their influences on gut microbiota.” presents an innovative approach to enhancing infant formula by incorporating dietary polysaccharides but would be strengthened by a deeper, more critical analysis of key elements such as mechanistic insights, safety profiles, and regulatory considerations. Including these aspects in a revised manuscript would offer a more comprehensive view of the practical challenges and impacts associated with polysaccharide supplementation in infant formula. Additionally, the complex bidirectional interactions between antibiotics and the gut microbiome (see: https://www.mdpi.com/2079-6382/12/9/1438) are a critical factor affecting infant gut health that warrants discussion. Addressing these interactions, alongside polysaccharide effects, would enhance understanding of how dietary polysaccharides could replicate the multifaceted bioactivity of human milk, ultimately guiding the development of more effective formula options. Additionally, the title could be simplified to highlight the key points of the review.

Author Response

The review titled: “A comprehensive review on dietary polysaccharides: As prebiotics, synbiotics, and postbiotics in infant formula and their influences on gut microbiota.” presents an innovative approach to enhancing infant formula by incorporating dietary polysaccharides but would be strengthened by a deeper, more critical analysis of key elements such as mechanistic insights, safety profiles, and regulatory considerations. Including these aspects in a revised manuscript would offer a more comprehensive view of the practical challenges and impacts associated with polysaccharide supplementation in infant formula. Additionally, the complex bidirectional interactions between antibiotics and the gut microbiome (see: https://www.mdpi.com/2079-6382/12/9/1438) are a critical factor affecting infant gut health that warrants discussion. Addressing these interactions, alongside polysaccharide effects, would enhance understanding of how dietary polysaccharides could replicate the multifaceted bioactivity of human milk, ultimately guiding the development of more effective formula options. Additionally, the title could be simplified to highlight the key points of the review.

Author response: Thanks for your comments. It really helps us to improve our manuscript. We have discussed about the safety and regulatory profiles, and these reveal that the applications of biotics in infant formula still need long way to go. Moreover, we also discussed the use of antibiotics to infants in their early life, and the possible effects on their gut microbiota. Breastfeeding may reduce the disruptions to the infant gut microbiome after antibiotic treated, while infant formula could increase the use of antibiotics. Therefore, we hope the future study could focus on the application of synbiotics/postbiotics to against the disruptions to infant microbiome in early life. Finally, we have revised the title of this review.